# A Structural In Silico Analysis of the Immunogenicity of L-Asparaginase from *Penicillium cerradense*

**DOI:** 10.3390/ijms25094788

**Published:** 2024-04-27

**Authors:** Kellen Cruvinel Rodrigues Andrade, Mauricio Homem-de-Mello, Julia Almeida Motta, Marina Guimarães Borges, Joel Antônio Cordeiro de Abreu, Paula Monteiro de Souza, Adalberto Pessoa, Georgios J. Pappas, Pérola de Oliveira Magalhães

**Affiliations:** 1Laboratory of Natural Products, Department of Pharmacy, Faculty of Health Sciences, University of Brasilia, Brasilia 70910-900, Brazil; kellen.cruvinel@gmail.com (K.C.R.A.);; 2inSiliTox, Department of Pharmacy, Faculty of Health Sciences, University of Brasilia, Brasilia 70910-900, Brazil; 3Department of Biochemical and Pharmaceutical Technology, School of Pharmaceutical Sciences, University of São Paulo, São Paulo 05508-000, Brazil; 4Department Cell Biology, Institute Biological Sciences, University of Brasilia, Brasilia 70910-900, Brazil

**Keywords:** L-asparaginase, immunogenicity, ALL, *Penicillium cerradense*

## Abstract

L-asparaginase is an essential drug used to treat acute lymphoid leukemia (ALL), a cancer of high prevalence in children. Several adverse reactions associated with L-asparaginase have been observed, mainly caused by immunogenicity and allergenicity. Some strategies have been adopted, such as searching for new microorganisms that produce the enzyme and applying protein engineering. Therefore, this work aimed to elucidate the molecular structure and predict the immunogenic profile of L-asparaginase from *Penicillium cerradense*, recently revealed as a new fungus of the genus *Penicillium* and producer of the enzyme, as a motivation to search for alternatives to bacterial L-asparaginase. In the evolutionary relationship, L-asparaginase from *P. cerradense* closely matches *Aspergillus* species. Using in silico tools, we characterized the enzyme as a protein fragment of 378 amino acids (39 kDa), including a signal peptide containing 17 amino acids, and the isoelectric point at 5.13. The oligomeric state was predicted to be a homotetramer. Also, this L-asparaginase presented a similar immunogenicity response (T- and B-cell epitopes) compared to *Escherichia coli* and *Dickeya chrysanthemi* enzymes. These results suggest a potentially useful L-asparaginase, with insights that can drive strategies to improve enzyme production.

## 1. Introduction

The utilization of microbial enzymes in biotechnological and therapeutic applications dates long back in science, and it is still a fertile ground with the prospects of exploring new activities from the biodiverse environment, aiming at improved scalability, kinetics, and safety, to list a few [1].

There is an ongoing quest to characterize the enzyme L-asparaginase (ASNase), EC 3.5.1.1, from different microorganisms, given its use in the food industry to prevent the formation of the carcinogenic by-product acrylamide, and mainly because of its use as an anti-tumor agent. The therapeutic use arises from the fact that some auxotrophic cancer cells, such as lymphoblasts in ALL, have diminished capacity to produce L-asparagine, relying upon scavenging it from the surroundings. The administration of ASNase is used to deplete the external pool of this amino acid, effectively causing disturbances in protein synthesis and leading to apoptosis [2,3].

ALL is a malignant neoplasm of B or T lymphoid progenitor cells with rapid progression, resulting from clonal proliferation and accumulation of cells that exhibit markers associated with early stages of lymphoid maturation [4,5]. It primarily affects children and young adults, with a significant occurrence between the ages of 2 and 5, and accounts for 75% of all leukemia diagnoses [6,7,8].

ASNase is used as a standard first-line option in remission-induction chemotherapy treatment for recent diagnoses and to direct prophylaxis of systemic chemotherapy for standard-risk and high-risk patients in ALL [9]. Thanks to advancements in innovation, efficacy, and access to treatments, the cure rate in children has increased to 90%. Additionally, the 5-year survival rate has also improved significantly, with over 80% of patients achieving disease-free survival, compared with 58% in 1970 [4,10,11]. 

ASNase catalyzes the hydrolysis of asparagine to aspartate and ammonium. This enzyme is widely distributed across all the kingdoms of living organisms and can be classified into different categories based on structural and biochemical criteria [12]. The prototypic enzymes used therapeutically are derived from *E. coli* and from *D. chrysanthemi* (syn. *Erwinia chrysanthemi*). Industrialized preparations of the enzyme are obtained from bacteria in three forms—native and polyethylene glycol-conjugated asparaginase (PEG-ASNase) from *E. coli*, and the native enzyme from *D. chrysanthemi* [2,13].

ASNase can be classified into three classes based on domains with distinctive structural and phylogenetic characteristics [12]: Class 1—InterPro ID: IPR027474; Class2—InterPro ID: IPR000246; and Class 3—InterPro ID: IPR010349. Each class is further divided into subgroups that have common functional, mechanistic, and structural characteristics [12]. The above-mentioned enzymes used for therapeutic purposes belong to Class 1; they are periplasmic and adopt a tetrameric catalytic state [12]. The WHO classified ASNase as a cytotoxic drug for ALL in the WHO Model List of Essential Medicines for Children and Adults [14,15]. Critical adverse reactions and toxicity associated with *E. coli* and *D. chrysanthemi* ASNase have been observed, such as hypersensitivity (clinical and subclinical), hypertriglyceridemia, encephalopathy, liver dysfunction, hyperglycemia, myelosuppression, pancreatitis, thrombosis, organ toxicity, glycosuria, ketoacidosis, decreased protein synthesis, hypoalbuminemia, and coagulopathies [7,16]. The most relevant toxic effects are related to the hypersensitivity mechanism and protein synthesis inhibition [17,18].

Strategies to reduce ASNase toxicity are needed to improve ALL therapy outcomes [19]. Several studies have aimed to search for new sources of ASNase since it can reduce the number and intensity of adverse effects [20,21]. Enzymes from eukaryotic origin, such as fungi, are attractive candidates on the premises of displaying distinct immune response elicitors to humans, compared with the standard bacterial sources [22,23].

Several studies have used bioinformatics tools to investigate structural, biochemical, and immunogenic characteristics of ASNase. These studies include structural investigations of ASNase from *Cavia porcellus* [24] and *Ocimum tenuiflorum* [25], heterologous expression of ASNase from *Aspergillus terreus* in *E. coli* [26], immunogenicity of ASNase from *E. coli* and *D. chrysanthemi* [27], and dehumanization of ASNase from *E. coli* and *Pectobacterium carotovorum* [28].

The present work aims to use computational tools to explore fungal alternatives to bacterial ASNase, centered on the investigation of the structure and immunogenic profile of the enzyme from *Penicillium cerradense*, a recently characterized fungus isolated from the Brazilian savannah [29].

## 2. Results

### 2.1. In Silico Analysis of P. cerradense L-Asparaginase Sequence Properties

The *P. cerradense* ASNase gene was identified from the genome sequence reported by our group in a previous study [29]. The protein sequence of ASNase from *P. cerradense* (NCBI accession: UDP03915.1) has 378 amino acids, a molecular mass of 39 kDa, and 5.13 isoelectric point, as predicted using ProtParam [30]. Protein domain prediction using InterProScan [31] confirms that the enzyme belongs to the asparaginase/glutaminase-like family (CDD id: cd00411) with a type II ASNase domain (IPR027474).

Prediction of signal peptides using SignalP [32] and Phobius [33] indicated a high probability of this export signal. The estimated half-life using ProtParam-expasy [30] was 30 h in mammalian reticulocytes. Calculations of the instability index (29.85) and the aliphatic index (97.25) indicated a stable protein [34]. 

Similarity searches using NCBI-BLASTp showed that the *P. cerradense* ASNase identity profile was as follows: *Schizosaccharomyces pombe* (NP_595021.1) 48.26% of identity; *Saccharomyces cerevisiae* (NP_013256.1), 47.73%; *Bacillus subtilis* (NP_388151.1), 48.17%; *E. coli* (NP_311860.1), 44.55%; and *E. coli* (NP_417432.1), 43.64%; all ASNase.

### 2.2. Phylogenetic Analysis of Fungal L-Asparaginases

The sequence identity of *P. cerradense*, when compared to clinically relevant bacterial ASNase, is 44.55% with *E. coli* (NP_311860.1) and 46.48% with *D. chrysantemi* (5F52_A). To find alternatives to these sources, a phylogenetic analysis was conducted using ASNase from several fungal genera. The maximum likelihood tree shown in Figure 1 reveals that *Penicillium* sp. ASNase forms a paraphyletic group with representatives from the *Aspergillus* genus (Figure 1), with the *P. cerradense* sequence placed in proximity to several *Aspergillus* sp. enzymes, particularly to *A. indologenicus* (PYI32151.1: 81.43% identity), than to the nearest *Penicillium* representatives (78.51% to *P. steckii* and 77.72% to *P. sizovae*). For instance, a sequence similarity of 56.99% was found between *P. cerradense* ASNase and *P. digitatum* (XP_014538187.1), a member of the divergent *Penicillium* clade in Figure 1.

### 2.3. Active Site Conservation

Representative ASNase sequences from Penicillium and Aspergillus species were analyzed focusing in two regions relevant to the enzymatic activity: the hinge region (HR) and the active-site flexible loop (ASFL). These regions, although not essential for substrate binding, exhibit conformational changes responsible for catalytic activity (conformations cat+ and cat−) after the substrate is bound [35]. In the ASNase, HR is a highly conserved glycine-rich octapeptide, with the canonical sequence GGTxyGGG (x = Ile or Leu; y = Ala or Gly). In *P. cerradense*, the HR presents the sequenceG^56^GTIAGSG^63^. The ASFL in *P. cerradense* presents the sequence S^64^SSTATTGYTAGAV^77^. However, there is no precise delimitation between the two regions. Among the fungal species under study, the HR region was highly conserved, with a difference only in the last two residues of this region (Figure 2). The ASFL region is usually seen as a variable region for ASNase from different microorganisms [36]. 

According to our analysis, *P. cerradense*, *P. sizovae*, and *P. steckii* ASFLs are closely related to the same region of the ASNase from *Aspergillus* genus yeasts. Other *Penicillium* species (*P. chrysogenum*, *P. digitatum*, *P. griseofulvum* and *P. italicum*) presented a different amino acid sequence pattern in the same region, but there were similarities among them. 

The active site of different ASNase was well preserved and rigid, with five critical residues for catalysis. These residues were a threonine in HR, a tyrosine in ASFL, and three other residues (ThrXXX-Asp and LysXXX) located about 64–67 and 137–140 residues apart, respectively, from the ASFL of the Tyr [37]. In *P. cerradense*’s ASNase these residues were Thr^58^ (HR), Tyr^72^ (ASFL), Thr^139^, Asp^140^, and Lys^212^. Data on polymeric interface and active site are the result of comparison with conserved residues and with crystallized ASNase structures.

The N-terminal portion showed low conservation among fungal ASNase (variable N-terminus), corresponding to the region of the first 38 residues for *P. cerradense*. This region was variable for the Penicillium and Aspergillus ASNase under study. 

### 2.4. Prediction of the Molecular Structure of L-Asparaginase

The *P. cerradense* ASNase three-dimensional model was predicted using AlphaFold2 [38,39] and is presented in Figure 3. The prediction confidence scores are presented in Appendix A, and are above 70% for most of the protein, with noticeable poor prediction scores in the N-terminus (residues 1–31). Ramachandran analysis showed 96.00% of its amino acid residues in regions favorable to the proposed model (Appendix A). 

Although phylogenetically distant, ASNase from *P. cerradense* presents close structural proximity to *E. coli* ASNase (PDB: 3ECA) and maintains an almost identical fold, with the presence of two α/β domains connected by a loop, displaying a Rossman fold topology (Appendix A) [40]. Figure 4 shows the structural similarity between ASNase from *P. cerradense*, *E. coli*, and *D. chrysantemi*. The comparison to the other bacterial commercially available ASNase (from *D. chrysantemi*, PDB: 5I4B) showed a closer structural similarity (1.40 Å and 47% sequence identity) than the enzyme from *E. coli* (1.51 Å and 43% sequence identity).

To examine the structural conservation of ASNase and gain insights into the sites important for catalysis and antigenicity, we expanded the structural investigation by generating AlphaFold2 predictions for all representatives of *Penicillium* and *Aspergillus* sp. used in our phylogenetic analysis (Section 2.2). Pairwise structure alignments (RCSB PDB) against the *P. cerradense* model presented structural proximity, with RMSD below 2.0 Å (Table 1). ASNase from *P. cerradense* is structurally closer to the enzyme from *A. indologenus* (0.7 Å and 79% sequence identity), whereas *P. sizovae* enzyme is the closest structure in the Penicillium genus (0.97 Å and 76% sequence identity).

The predicted models for *Penicillium* and *Aspergillus* ASNase were expected to have a close structural similarity, as their sequence similarity was found to be above 45.31%. However, sequence conservation varies and is higher between residues 49–236, corresponding to the ASNase N-terminal conserved domain (InterPro: IPR027474). This is particularly evident in the catalytic important regions HR and ASFL (Figure 2). By projecting the amino acid conservation values onto the *P. cerradense* model structure and dividing the structure with a frontal plane, two sections with distinct conservation arise: the more conserved ventral face, which contains the active site, and the less conserved dorsal face (Figure 5).

This high conservation in the catalytic site of ASNase may suggest similar enzymatic kinetics for the genera *Penicillium* and *Aspergillus*, whereas the more divergent dorsal face may produce distinct immune responses depending on the ASNase species.

AlphaFold2 modeling did not show the possible final oligomeric conformation of the protein. Thus, for the prediction of the quaternary structure, the enzyme was modeled using SwissModel–Expasy, by homology. The sequence was predicted as a homotetramer (Quaternary Structure Quality Estimate—QSQE 0.81), 46.30% aligned to the template sequence obtained from *Dickeya chrysanthemi* (PDB 5I4B). The global model quality estimate (GMQE) was calculated as 0.71 and the qualitative model energy analysis with distance constraints (QMEANDisCo Global) was 0.77 ± 0.05. The modeled structure shows conformity with previous reports of the tetrameric structure of ASNase [40]. The enzyme ASNase is commonly found as a tetramer, but monomeric, dimeric, and hexameric forms have also been found in isolates from different sources [3]. In any case, other sources and post-translational modifications can strongly influence the enzyme’s molecular structure [41].

### 2.5. Prediction of Immunogenicity, Allergenicity and Toxicity

The prediction of T-cell epitope density allows the inference of the degree of immunogenicity (DI) [42,43,44]. In this regard, evaluating the immunogenicity of proteins of therapeutic value has commonly used the density of epitopes as an indicator [45,46,47].

This concept was used to assess the immunogenicity of the *P. cerradense* ASNase compared with available clinical-use ASNase (*E. coli* and *D. chrysanthemi*) and ASNases from other *Penicillium* and *Aspergillus* species. The density of epitopes and the DI were evaluated using eight globally distributed alleles as a reference (HLA-DRB1*01:01, HLA-DRB1*03:01, HLA-DRB1*04:01, HLA-DRB1*07:01, HLA-DRB1*08:01, HLA-DRB1*11:01, HLA-DRB1*13:01, and HLA-DRB1*15:01) [48,49].

The DI of *P. cerradense*’s ASNase showed no significant difference compared with the enzymes of *E. coli*, *D. chrysanthemi*, *P. chrysogenum*, *P. digitatum*, *P. griseofulvum*, *P. italicum*, *P. sizovae*, *P. steckii*, *A. ibericus*, *A. idologenus*, *A. niger*, and *A. sclerotiicarbonarius* (Figure 6). In comparison with the clinical ASNase, *P. cerradense* protein showed a higher DI (0.0149) than *E. coli* (0.0142) and lower than the enzyme from *D. chrysanthemi* (0.0197). The ASNase of *D. chrysanthemi* presented the highest DI. The results obtained are similar to those reported by Belen et al. (2019), that the ASNase DI from *D. chrysanthemi* was higher than the ASNase DI from *E. coli* [27]. 

Evaluating the DI, it was verified that there was no significant difference in the predicted immunogenicity among the ASNase of the microorganisms considered in this study. However, when comparing the relative frequencies of fungal ASNase with those of clinical use from *E. coli* and *D. chrysanthemi*, the predicted DI for fungal ASNase was equivalent to or lower than that of bacterial enzymes.

Among *Penicillium* species, ASNase DI values exhibited similar epitope density: *P. cerradense* (0.0149), *P. chrysogenum* (0.0147), *P. digitatum* (0.0150), *P. griseofulvum* (0.0144), and *P. italicum* (0.0142), whereas *P. sizovae* and *P. steckii* ASNase presented values of 0.0132 and 0.0119, respectively, which were the closest to the value that was computed for *E. coli* (0.0142). Analyzing the *Aspergillus* species, DI values exhibited the largest variation in epitope density: *A. ibericus* (0.0158), *A. sclerotiicarbonarius* (0.0161), *A. indologenus* (0.0135), and *A. niger* (0.0141). Considering the DI by genus (*Penicillium* and *Aspergillus*), the results presented may suggest a trend of DI by genus, but at the same time, possible particularities of each species can be perceived.

Figure 7 represents the T-cell epitope density for each of the eight alleles (HLA-DRB1*01:01, HLA-DRB1*03:01, HLA-DRB1*04:01, HLA-DRB1*07:01, HLA-DRB1*08:01, HLA-DRB1*11:01, HLA-DRB1*13:01, and HLA-DRB1*15:01). The predicted immunogenic T-cell epitopes are presented in Appendix A. The results show heterogeneity in distribution for each species depending on the allele. Compared with the enzymes of clinical use (*E. coli* and *D. chrysanthemi*), ASNase from *P. cerradense* showed a similar or lower density value of the eight evaluated alleles. In general, similar distribution behavior can be observed with the trends for each allele for the ASNase evaluated.

Specifically, the HLADRB1*07:01 allele is well recognized as being associated with a high risk of hypersensitivity reactions and a higher risk of allergies after treatment with bacterial ASNase, possibly because it is an allele that confers high-affinity binding [50,51]. Based on these assumptions, the epitope density and prediction of ASNase allergenic peptides from *P. cerradense* for the HLA-DRB1*07:01 allele were compared with the same results for the native enzymes from *E. coli*, *D. chrysanthemi*, and species of the genera *Penicillium* and *Aspergillus* to determine the amino acids and regions that could contribute to allergenicity. Figure 8 presents the epitope density of allergenic T-cell epitopes for the HLADRB1*07:01 allele of ASNase from *P. cerradense* and other microorganisms in comparison.

ASNase from *P. cerradense* showed the density of allergenic epitopes (0.603) larger than the enzyme from *E. coli* (0.526) and lower than the enzyme from *D. chrysanthemi* (0.653). The ASNase of *D. chrysanthemi* presented the highest density of allergenic epitopes (0.653), followed by *P. digitatum* (0.647), *P. italicum* (0.634), *A. idologenus* (0.632), and *A. sclerotiicarbonarius* (0.625). Conversely, the ASNase of *P. steckii* and *P. sizovae* presented the lowest densities of allergenic epitopes (0.450 and 0.492, respectively). 

The mapping of the allergenic peptide fragments in the ASNase structures showed a different distribution profile for the enzyme from *P. cerradense* compared with *E. coli* and *D. chrysanthemi* (Figure 9). The structural determinants for providing the allergenicity characteristics are summarized in Appendix A. Despite the difference in distribution, the ASNase from *P. cerradense, E. coli*, and *D. chrysanthemi* showed similar structural regions in concentration of allergenic epitopes (Figure 10). There are six regions: chain Nβ1/HR, Ncoil7, helix Nα4, chain Nβ6, linker interdomain, and helix Cα2, respectively in the I^52^FGTGGTIA^60^, M^165^RPSTATSA^173^, F^177^NLLEAVTV^192^, Y^208^YVTKTNAN^218^, F^252^DITATKEI^260^, and F^298^NHAIEDVI^310^ positions in the ASNase model *P. cerradense*. The epitopes concentrated in these regions showed divergence in terms of amino acid sequence despite the same spatial location, which suggests that these are allergenic regions mainly determined by spatial conformation and surface exposure.

Linear B-cell epitope prediction was performed to evaluate the enzyme’s ability to generate antibodies. Linear B-cell epitope diagnosis was performed for ASNase from *P. cerradense* and compared with *E. coli*, *D. chrysanthemi*, and species of the genera *Penicillium* and *Aspergillus to* determine the amino acids and regions that could contribute to generate antibodies. Figure 11 represents the B-cell DI for this comparison. Appendix A presents the linear B-cell epitopes identified in the amino acid sequence of these ASNase.

The *P. cerradense*’s ASNase showed a similar DI (0.069) to *D. chrysanthemi* (0.061), more than 40% lower than the enzyme from *E. coli* (0.117). The ASNase from *E. coli* presented the highest DI. This result is congruent with reports of increased hypersensitivity and antibody formation by *E. coli* ASNase (60% of patients under treatment), more than reported under *D. chrysanthemi* (8–33% of patients under treatment), which has been described as immunogenically distinct [52,53,54].

Comparing fungal ASNase with those in clinical use produced by *E. coli* and *D. chrysanthemi*, the predicted B-cell DIs for fungal ASNase were equivalent to or smaller than those of bacterial enzymes. Analyzing the *Penicillium* species, DI values displayed variation in relative frequency: *P. cerradense* (0.069), *P. chrysogenum* (0.082), *P. digitatum* (0.071), *P. griseofulvum* (0.092), and *P. italicum* (0.071), *P. steckii* (0.065), whereas *P. sizovae* (0.052) ASNase presented a DI closest to the value that was computed for *D. chrysanthemi* (0.061).

Among *Aspergillus* species, B-cell DI values exhibited similar relative frequencies: *A. ibericus* (0.037), *A. idologenus* (0.032), *A. niger* (0.042), and *A. sclerotiicarbonarius* (0.048). All species of the genus presented the lowest B-cell DI among the microorganisms evaluated in this study. The presented results may suggest that fungal ASNase from *Aspergillus* is less able to generate antibodies through this pathway.

The mapping of B-cell epitopes on monomeric structures was performed for ASNase from *P. cerradense* and compared with *E. coli* and *D. chrysanthemi* enzymes (Figure 12). The structural distribution showed a different profile for ASNase from *P. cerradense*, *E. coli*, and *D. chrysanthemi*. 

The length (predicted number of epitopes) of the immunogenic residues of *P. cerradense* is dissimilar to those from *E. coli* and *D. chrysanthemi*. In the immune response to B-cell epitopes, ASNase from *P. cerradense* may present itself in a distinct immunogenic manner. 

Regarding the toxicity profile, from analysis on the ToxinPred server, interestingly, no toxic peptide fragments were found in the sequence of ASNase from *P. cerradense*, a result indicative of a non-toxic protein. In contrast, *E. coli* ASNase has been reported to have a highly toxic region responsible for its adverse effects [55]. Among all studied species, only *P. italicum* and *P. sizovae* presented toxic peptide fragments in their ASNase sequences. 

Using computational tools, the antigenic structural determination that may contribute to the generation of hypersensitivity response associated with ASNase from *P. cerradense* was determined for the first time in the present study. The obtained results need still to be confirmed through clinical or laboratorial validation of the immunogenic and allergenic epitopes that would corroborate the in silico predicted activity. The potential of the computational analysis achieved for ASNase from *P. cerradense* could be validated by generating mutants and evaluating their ability to elicit hypersensitivity reactions, considering that a decrease in responses should be expected after the intervention of the identified epitopes.

## 3. Discussion

The sequence of the *P. cerradense* ASNase gene, identified from its complete genomic sequence, has 1251 pb. The predicted protein showed homology to other ASNase from *Aspergillus* genus. Among ASNases from *Penicillium* genus, it was closer to *P. sizovae* and *P. steckii* ASNase. These two *Penicillium* species are phylogenetically close to *P. cerradense* and belong to the same—citrine section—group [29].

The evolutionary relationships of ASNase gene differ among species-level trees, with inconsistencies. This conflict can occur due to incomplete lineage sorting (ILS) and/or introgression by hybridization [56]. The ASNase from different species of *Penicillium* may present distinct or discrepant functional behaviors, and/or similar behaviors to the species of the genus *Aspergillus*. To the extent of our knowledge, this is the first time that the evolutionary relationship for ASNase from *Penicillium* has been reported in the literature. Differences in the gene region of ASNase may represent complexity and difficulty in comparison studies using amino acid sequences. Likewise, this difference may suggest a wide variety of the enzyme in regard to its immunogenicity, specificity, and stability, according to the source [57].

From the sequence and structural analyses, it can be inferred that ASNase from *P. cerradense* has type II asparaginase activity (periplasmic, with 17 aa signal peptide) with 5.13 pI, and that it is a tetramer with an estimated theoretical molecular mass activity of 156 kDa (39 kDa per monomer), similar to those purified from *Citrobacter* (166 kDa) [58] and *E. carotovora* (130/152 kDa) [59]. With the pairwise alignment of the ASNases, it was possible to compare the structure of *P. cerradense* ASNase with other, evolutionary distant, ASNase. Remarkable structural similarity among ASNase monomers of microorganisms was observed. This similarity relationship is in line with what was reported in the study by da Silva et al. (2022) where they indicated that class 1 ASNase showed high conservation in the tertiary structure even with low amino acid identity, suggesting a common evolutionary ancestry [12]. When comparing the ASNase from *P. cerradense* with the two clinically approved ASNases, a higher structural similarity is presented to the *D. chrysanthemi* enzyme. This similarity may suggest a closer functional behavior to *D. chrysanthemi* ASNase than to *E. coli*. For the fungi under study, the structural similarity is consistent with the characteristic of a grouping of the evolutionary relationship for the enzyme ASNase and the formation of a paraphyletic group.

The epitope density was applied to evaluate the immunogenicity of the enzyme ASNase from *P. cerradense*, related to T-cell and B-cell responses, and, in parallel, the study evaluated allergenicity and toxicity in comparison to commercially available ASNase. ASNase from *P. cerradense* showed similar or less T-cell/B-cell immunogenicity compared with the *E. coli* and *D. chrysanthemi* ASNase. While the density of epitopes presented was close, the structural distribution of the allergenic epitopes of the ASNase from *P. cerradense* was not the same when compared with the enzymes produced by *E. coli* and *D. chrysanthemi*, which could predispose to a different pattern of response to allergenicity. It might be possible to infer that *Penicillium cerradense’s* ASNase immunologic safety required for clinical use is similar to those already marketed. However, in vitro and/or in vivo evaluation is needed to confirm these assumptions. The results of this analysis can help in the biotechnological improvement of new fungal ASNases like the one presented in this study from *P. cerradense*, if they show promising clinical responses. To its advantage, the ASNase from *P. cerradense* presents itself as a non-toxic protein, different from *E. coli*’s that contains a highly toxic region responsible partly for its adverse effects [55].

This study shows that there were no significant differences in the level of immunogenicity between fungal and bacterial ASNase in the studied species. In contrast, bacterial ASNase from *D. chysanthemi* showed a higher relative frequency of T-cell and allergenic epitopes, and *E. coli* showed a higher relative frequency of B-cell epitopes. *Penicillium* and *Aspergillus* ASNase presented a degree of immunogenicity compatible with clinical use, and this study observed that immunogenicity seems to be associated with the species but presents an individuality of behavior for each enzyme. Among the studied enzymes, the best results regarding the prediction of the immunogenic response were achieved by *P. steckii* ASNase.

Hypersensitivity reactions caused by ASNase from *E. coli* have been widely studied in vivo [60,61,62]. However, there has been too little research in this area relating to fungal ASNase. The data presented in this study may help further research on the clinical use of fungal ASNase and provide a solution with less immunogenic response. Evaluating the general results of hypersensitivity, fungal ASNase has the potential to be applied as a clinical first choice, especially replacing the enzyme produced by *D. chysanthemi.*

The data obtained in this work present the properties of *P. cerradense* ASNase and predict its possible cellular immune responses, improving the understanding of its molecular structure, and complementing the initial research about the production of this enzyme previously presented by our group [29]. Overall, the data presented are relevant to pave the way toward the understanding of the functional behaviors of ASNase from the genus *Penicillium*, in comparison with commercially available ASNase used in the treatment of ALL *(E. coli* and *D. chrysanthemi*). The results of these predictions will help to develop strategies to reduce adverse immune responses to this enzyme. Techniques based on structural modifications may bring answers and more effective approaches to ASNase enzyme treatments.

## 4. Materials and Methods

### 4.1. Microorganisms and L-Asparaginase Gene Sequences

Previously, we isolated a fungus from the Brazilian savannah soil and identified it as a new species of the genus *Penicillium*. This fungus, *P. cerradense*, was identified as a producer of ASNase [29]. The enzyme gene sequence deposited under accession code MT742156 in NCBI GenBank was used in the present research.

Amino acid structures and sequences of *E. coli* (3ECA) and *D. chrysanthemi* (2JK0) ASNase obtained from the Protein Data Bank (PDB) were used as the standard for the clinically available enzymes. The NCBI accession codes of other *Penicillium* species used in the present work as sources of ASNase were *P. chrysogenum* (XP_002563013), *P. digitatum* (XP_014538187.1), *P. griseofulvum* (KXG45967.1), *P. italicum* (KGO77393.1), *P. steckii* (OQE28485.1), and *P. sizovae* (MW291568). After the phylogenetic results, we added ASNase of species from the *Aspergillus* genus because of their evolutionary proximity with *P. cerradense*. The species and NCBI accession codes were *A. idologenus* (PYI32151.1), *A. ibericus* (XP_025570260.1), *A. sclerotiicarbonarius* (PYI04731.1), and *A. niger (XP_001389884.1*). Some of the selected species that showed experimental ASNase activity (*P. chrysogenum*, *P. digitatum*, *P. sizovae* and *A. niger* [17,63,64,65]), while the others (*P. griseofulvum* [66], *P. italicum* [67], *P. steckii* [68], *A. idologenus*, *A. ibericus* and *A. sclerotiicarbonarius* [69]) had their ASNase sequence identified through their genome studies. All sequences are listed in Appendix A.

### 4.2. In Silico Analysis of L-Asparaginase from P. cerradense

The obtained nucleotide sequence was translated using the Expasy translation tool and analyzed through database sequences using NCBI BLAST. The deduced amino acid sequence’s molecular masses, theoretical pI values, and other physicochemical properties were predicted using Expasy’s ProtParam tool (https://web.expasy.org/protparam/ accessed on 1 August 2023). SignaIP 5.0 [32] was used to predict the presence of any signal peptide in the translated sequence. The result was confirmed using Phobius (https://phobius.sbc.su.se/ accessed on 1 August 2023) [33]. Evolutionary analysis of the proteins was performed from 35 protein representatives with ASNase predicted activity that had some degree of similarity to the ASNase from *P. cerradense* obtained from the GenBanK/PSIBlast. Sequence alignment was performed considering the conserved domain of ASNase through the IQ-TREE (version 2.2.0) [70] software using the maximum likelihood method, using the Q.yeast+R3 model with 1000 bootstrap replicates. *E. coli* (ETI79984.1) was chosen as an outgroup for the analysis. In addition, to determine the type of ASNase (I or II) from *P. cerradense*, the NCBI’s conserved domain database (CDD) and the EMBL-EBI InterPro tool (https://www.ebi.ac.uk/interpro/ accessed on 1 August 2023) were used for superfamily and protein domain analysis. Functional sites and motifs were inspected with Prosite/ExPASy (https://prosite.expasy.org/ accessed on 1 August 2023). The hinge region (HR) and active-site flexible loop (ASFL), essential regions to the stabilization of the catalytic site [37], were aligned and compared.

### 4.3. Prediction of the Molecular Structure and Insights of L-Asparaginase from P. cerradense

Molecular modeling of the three-dimensional structure was performed using AlphaFold2 software v1.5.5 [38,39] (DeepMind/EMBL accessed on 1 July 2023). The reliability of the proposed three-dimensional homology model was evaluated via Ramachandran analysis using Verify3D accessed on July 2023 [71].

Molecular prediction of the oligomeric structure was performed using SWISS-MODEL software (https://swissmodel.expasy.org/ accessed on 1 June 2023). The parameters for reliability of the in silico evaluation were GMQE (global model quality estimation) [72], QME-ANDisCo (qualitative model energy analysis with distance constraints) [73], and QSQE (quaternary structure quality estimate) [74] in SWISS-MODEL.

### 4.4. Pairwise Structural Alignments L-Asparaginases

For structural comparison between the enzyme ASNase from *P. cerradense* and the sources of ASNase available for clinical use, the files of protein structures from *E. coli* (3ECA) and *D. chrysanthemi* (2JK0) obtained from the Protein Data Bank (PDB) were used. Pairwise structure alignment was performed using the web service available at RCSB PDB (Research Collaboratory for Structural Bioinformatics PDB) (https://www.rcsb.org/alignment accessed on 1 July 2023) with the jFATCAT—flexible algorithm. Alignment evaluation parameters were RMSD (root-mean-square deviation) and TM score with scores between 0 and 1, where values > 0.5 represent models with the same protein fold.

### 4.5. Sequence-Structure Conservation of Fungal L-Asparaginases

Multiple sequence alignment using 3D structure support was performed using the T-Coffee Expresso web server (https://tcoffee.crg.eu/apps/tcoffee/do:expresso accessed on 1 April 2023). *P. cerradense* ASNase was used as the template for the alignment and superposition of the other species’ enzymes. Mapping of structure sequences’ conservation was performed using the UCSF Chimera program. Visualization of sequence conservation onto molecular structures was performed using the UCSF Chimera program [75].

### 4.6. Prediction of Epitopes in T-Cells and Determination of Epitopes Density

The MHC-II binding predictions from the Immune Epitope Database (IEDB) (http://tools.iedb.org/mhcii/) were used for T-cell epitope prediction. The IEDB-recommended (2023.05—NetMHCIIPan 4.1 EL) method for the program was selected [76]. The method used was NetMHCIIPan 4.1 EL, consisting of a neural network that predicted the MHC binding values from an amino acid sequence, based on a training set of peptide-MHC class II quantitative binding data covering thousands of human MHC molecules, including HLA-A, HLA-B, HLA-C, HLA-E, and HLA-G [76].

Peptides predicted as immunogenic epitopes were linearly established by percentile < 2 (“strong binder”) and < 10 (“weak binder”). For epitope density determination, the relative frequency calculation (Equation (1)) was used:f_i_ = n_i_/N = n_i_/(Σ_jnj_)(1)
where n_i_ = number of predicted immunogenic epitopes, N = total number of epitopes predicted by the program, and Σ_jnj_ = epitopes predicted immunogenic and non-immunogenic.

The epitope density of each protein was determined for the alleles HLA-DRB1*01:01, HLA-DRB1*03:01, HLA-DRB1*04:01, HLA-DRB1*07:01, HLA-DRB1*08:01, HLA-DRB1*11:01, HLA-DRB1*13:01, and HLA-DRB1*15:01. These alleles were selected for prediction with reference to their high global distribution [48,49] and conferral of high-affinity binding to ASNase epitopes, causing various immunogenic reactions [50]. The prediction of T-cell epitope density allows the inference of the degree of immunogenicity (DI) [42,43,44]. This measurement alternative conceptualizes and compares a more immunogenic protein with another, determining whether the epitope density is greater. In this regard, evaluating the immunogenicity of proteins of therapeutic value has commonly used the density of epitopes as an indicator, inferring the degree of immunogenicity [45,46,47].

### 4.7. Prediction of T-Cell Epitopes Allergenicity

The prediction of the allergenicity that can be induced by the ASNase from *P. cerradense*, *E. coli*, and *D. chrysanthemi* was performed using the software AllerTOP v.2.0 (https://www.ddg-pharmfac.net/AllerTOP/ accessed on 1 July 2023) [77]. This software evaluated each of the T-cell epitopes predicted to be immunogenic for the HLA-DRB1*07:01 allele. The possible endpoints provided were “probable allergen” and “likely non-allergen”. The relative frequency of allergen epitopes was calculated by dividing the number of immunogenic epitopes for the HLA-DRB1*07:01 allele over the total number of immunogenic epitopes previously determined for this allele, as described for each ASNase in the previous section. 

### 4.8. Prediction of Epitopes in B-Cells and Toxicity

The prediction of B-cell epitopes (conformational epitopes) was performed using the BepiPred-3.0 server: (https://services.healthtech.dtu.dk/services/BepiPred-3.0/ accessed on 1 July 2023) [78] from the DTU Health Tech resource database. To assess toxicity, the webserver ToxinPred (http://crdd.osdd.net/raghava/toxinpred/ accessed on 1 July 2023) [79] was used to predict toxic/non-toxic peptides.

### 4.9. Epitope Mapping

Epitope mapping was performed in monomeric structures modeled of ASNase with Dassault Systèmes BIOVIA (https://www.3ds.com/products-services/biovia/ accessed July 2023). BIOVIA Discovery Studio v. 2021 was used to generate the graphical results [80].

### 4.10. Statistical Analysis

GraphPad Prism® Version 6.01 software was used for statistical analysis. The distribution of the data was evaluated and non-parametric tests were applied (data represented by median and interquartile ranges). The tests used for each experiment analysis are described in the legends of the representative graphs. Significant difference was considered at *p*-value < 0.05.

## 5. Conclusions

In silico analysis combined with bioinformatics tools revealed enzyme properties and predicted the immune responses that might arise from *P. cerradense* ASNase. This recently described enzyme shows a similar immunogenic and allergenic pattern compared with the ASNase already in clinical use. At the same time, it was predicted as a non-toxic protein. These results may drive strategies to improve the production of this enzyme and lead to potential production with desirable functional characteristics for enhanced therapeutic applications in ALL.

## Figures and Tables

**Figure 1 ijms-25-04788-f001:**
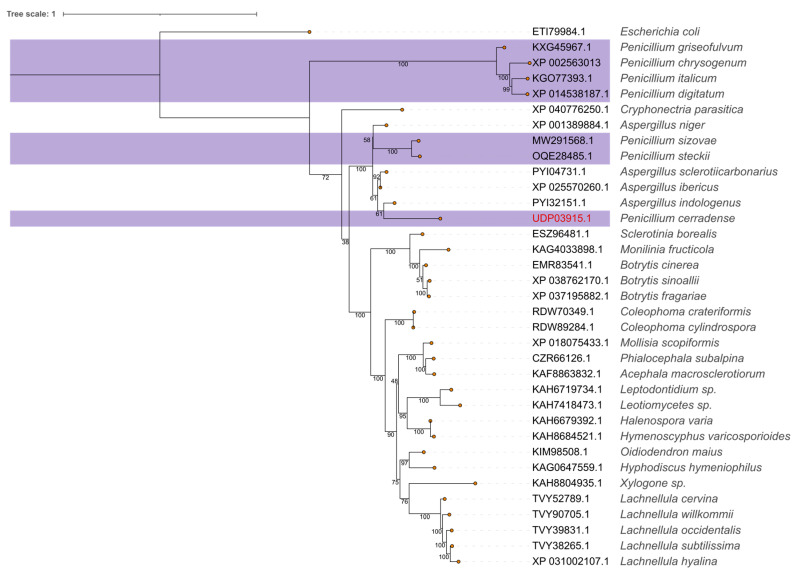
Maximum likelihood phylogenetic tree of L-asparaginases from several fungal species including *P. cerradense*. The consensus tree was inferred using IQ-TREE with 1000 ultrafast bootstrap replicates. The tree was rooted using the *E. coli* sequence as the outgroup and bootstrap support values are indicated below node edges. The *Penicillium* sp. representatives are highlighted.

**Figure 2 ijms-25-04788-f002:**
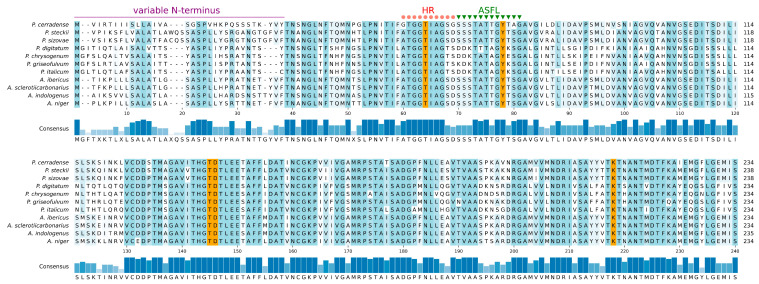
Hinge region (HR), active-site llexible loop (ASFL), and active-site residues’ alignment. Alignment of the Penicillium and Aspergillus L-asparaginases including *P. cerradense*. HR is indicated in red, ASFL in green, the residues relevant to the catalytic activity are highlighted in orange, and variable N-terminus. Consensus is the identity percentage, and full alignment residue differences are shown (in blue) among the compared asparaginases.

**Figure 3 ijms-25-04788-f003:**
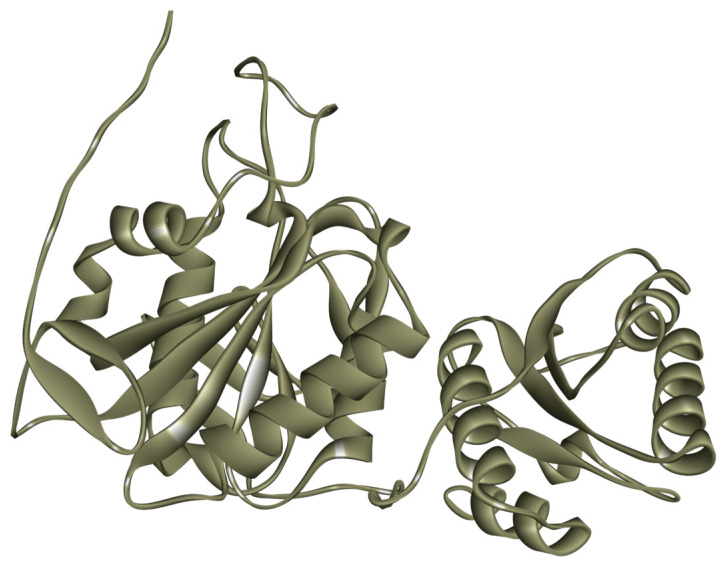
Model of the monomeric three-dimensional structure of L-asparaginase from *P. cerradense* obtained using AlphaFold2 v1.5.5.

**Figure 4 ijms-25-04788-f004:**
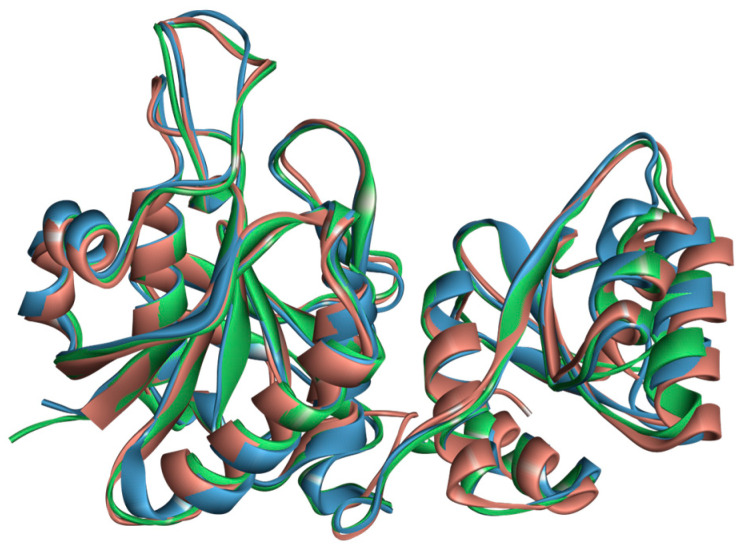
Structural overlap of L-asparaginase from *P. cerradense* (red) to L-asparaginase from *E. coli* (PDB: 3ECA) (blue) and *D. chrysanthemi* (PDB: 2JK0) (green).

**Figure 5 ijms-25-04788-f005:**
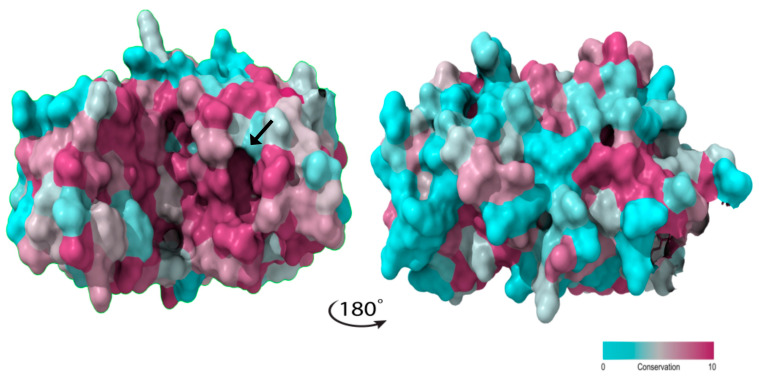
Sequence-structure conservation mapping for L-asparaginases from *Penicillium* and *Aspergillus* genera (used in Figure 2), using the ASNase from *P. cerradense* as a model. Catalytic cavity is indicated with a black arrow. The figure was generated using the ConSurf server v.3.

**Figure 6 ijms-25-04788-f006:**
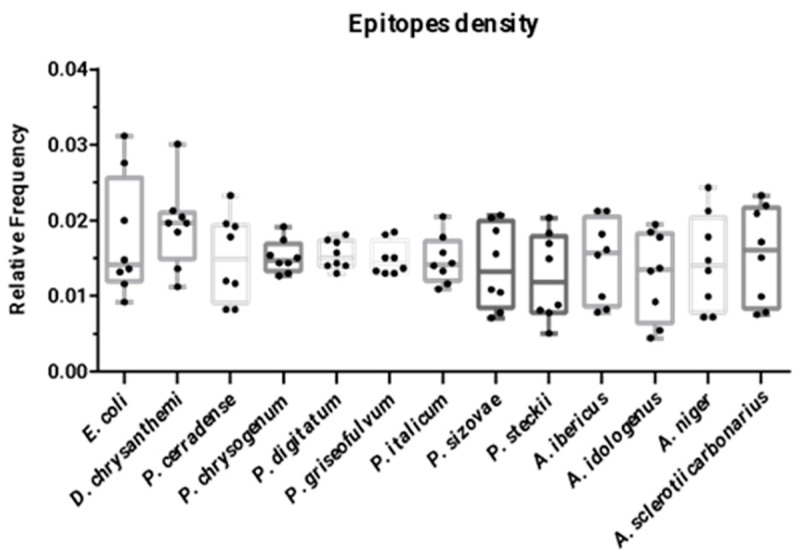
Dot plot graph (median and interquartile ranges) representing the immunogenicity degree of the L-asparaginases by epitope density predicted for T-cell immunogenic epitopes for eight alleles (HLA-DRB1*01:01, HLA-DRB1*03:01, HLA-DRB1*04:01, HLA-DRB1*07:01, HLA-DRB1*08:01, HLA-DRB1*11:01, HLA-DRB1*13:01, and HLA-DRB1*15:01). The dots in the box represent the eight alleles evaluated. No statistically significant difference was observed (*p* < 0.05—Kruskal–Wallis with a posteriori Dunn’s test).

**Figure 7 ijms-25-04788-f007:**
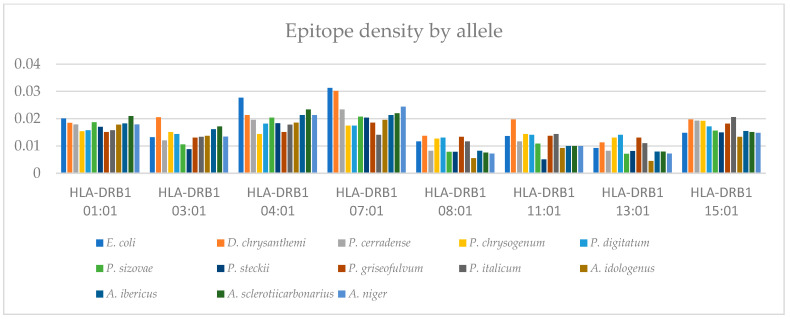
Bar graph representing the epitope density of different L-asparaginases, via predicting T-cell immunogenic epitopes for eight independent alleles.

**Figure 8 ijms-25-04788-f008:**
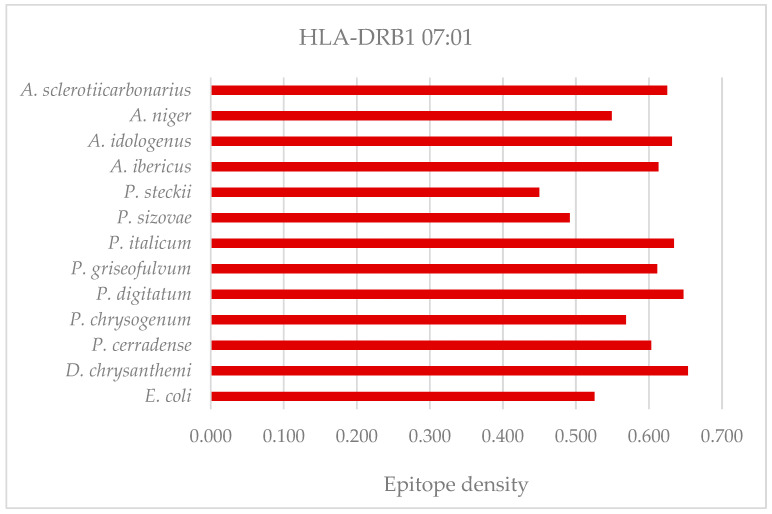
Epitope density of T-cell allergenic epitopes for the HLA-DRB1*07:01 allele of L-asparaginase from the evaluated microorganisms.

**Figure 9 ijms-25-04788-f009:**
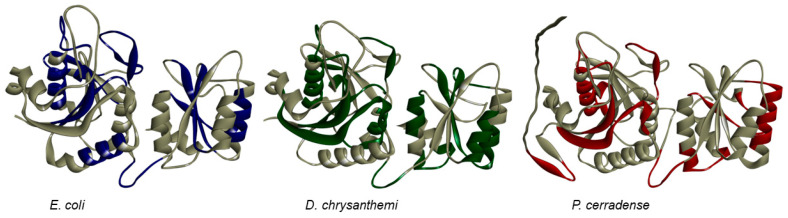
Structural distribution of T-cell allergen epitopes for the HLA-DRB1*07:01 allele in the L-asparaginase monomer. Blue zones represent *E. coli* allergenic epitopes. Green zones represent *D. chrysanthemi* allergenic epitopes. Red zones represent *P. cerradense* allergenic epitopes. Gray zones represent non-allergenic zones.

**Figure 10 ijms-25-04788-f010:**
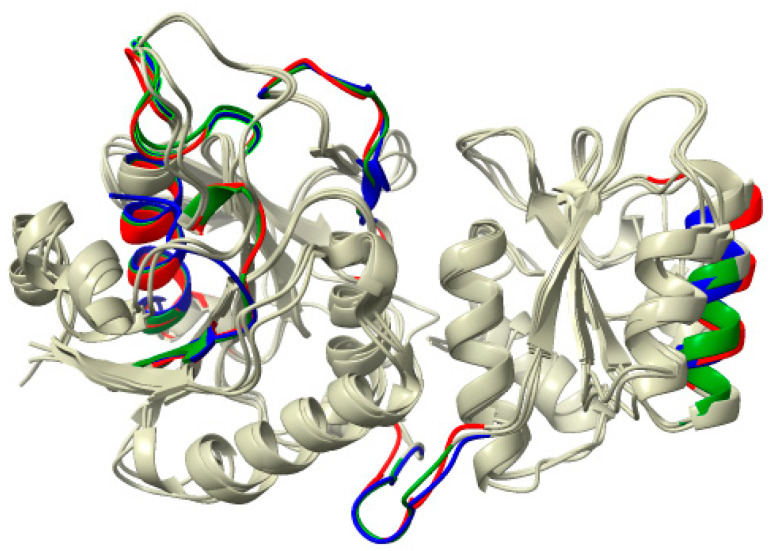
Structural conservation mapping of T-cell allergen epitopes for the HLA-DRB1*07:01 allele in the L-asparaginases of *P. cerradense, E. coli*, and *D. chrysanthemi.* Gray zones represent non-allergenic regions. Blue zones represent *E. coli* allergenic epitopes. Green zones represent *D. chrysanthemi* allergenic epitopes. Red zones represent *P. cerradense* allergenic epitopes.

**Figure 11 ijms-25-04788-f011:**
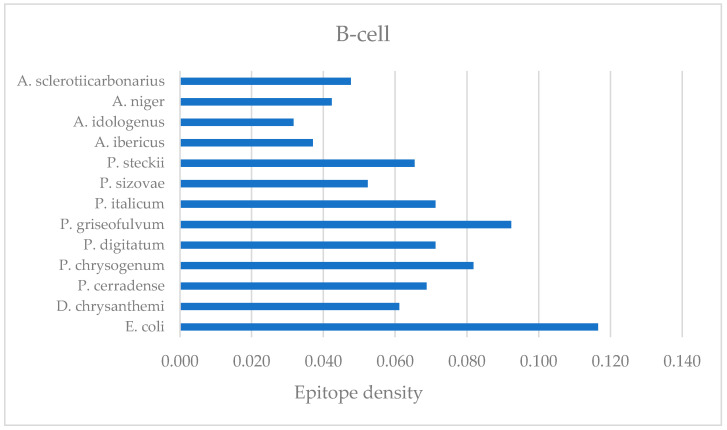
Bar graph representing the epitope density of B-cell epitopes of L-asparaginase from the evaluated microorganisms.

**Figure 12 ijms-25-04788-f012:**
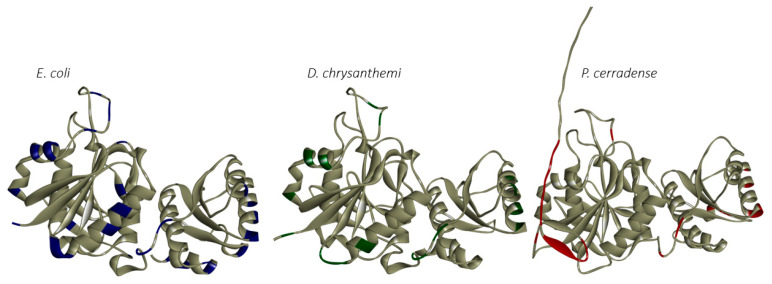
Structural distribution of B-cell epitopes for L-asparaginase monomers. Blue zones represent *E. coli* epitopes. Green zones represent *D. chrysanthemi* epitopes. Red zones represent *P. cerradense* epitopes. Gray zones represent non-allergenic zones.

**Table 1 ijms-25-04788-t001:** Pairwise structure alignment of *P. cerradense* AlphaFold2 structural model against other predictions of fungal L-asparaginases. The X-ray experimental structures from bacteria (*E. coli* and *D. chrysanthemi*) are also shown. Alignment evaluation parameters were RMSD (root mean square deviation) and TM score (template modeling score).

Microorganisms	NCBI id	RMSD	TM-Score	Sequence Identity	Equivalent Residues
*E. coli*	3ECA_A	1.51	0.82	43%	322
*D. chrysanthemi*	2JK0_A	1.40	0.82	47%	322
*P. chrysogenum*	XP_002563013	1.23	0.40	54%	377
*P. digitatum*	XP_014538187.1	1.11	0.38	57%	375
*P. griseofulvum*	KXG45967.1	1.02	0.38	57%	375
*P. italicum*	KGO77393.1	0.95	0.65	57%	364
*P. steckii*	OQE28485.1	1.03	0.67	77%	377
*P. sizovae*	MW291568	0.97	0.69	76%	377
*A. ibericus*	XP_025570260.1	0.98	0.41	79%	373
*A. indologenus*	PYI32151.1	0.70	0.71	79%	374
*A. niger*	XP 001389884.1	0.85	0.59	80%	369
*A. sclerotiicarbonarius*	PYI04731.1	1.12	0.31	79%	375

## Data Availability

Data is contained within the article and Appendix A.

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
