# Peer review of "A Structural In Silico Analysis of the Immunogenicity of L-Asparaginase from Penicillium cerradense"

_ijms, 2024, doi:10.3390/ijms25094788_

Round 1
Reviewer 1 Report
Comments and Suggestions for Authors
The article “A structural in silico analysis of the immunogenicity of L-asparaginase from Penicillium cerradense” has high applied significance, because algorithms for analyzing proteins and peptides for their immunogenicity and allergenicity have not yet been fully studied.
Key notes:
1. It is not clear why the structures of L-asparaginase were not relaxed by molecular dynamics. It is known that pdb structures are the result of processing X-ray structural analysis data and protein structures were obtained in vacuum. For most in silico work, pdb structures are recommended to be processed by molecular dynamics.
2. The authors should describe in more detail the algorithms for identifying epitopes in association with alleles.
3. It is also necessary to disclose in more detail the algorithm that made it possible to obtain specific sequences on lines 310-312.
Technical Notes:
I recommend combining the Supplementaries into one file.
Lines 2 and 585: in silico should be italicized.
Line 3: Penicillium cerradense should be italicized.
The caption for Figure 1 and Figure 1 itself should be placed on the same page.
Figure 7 appears to have part of the design cut off at the bottom.
Author Response
Dear Editors and Reviewers,
Thank you for considering the manuscript as potentially valuable for publication. On behalf of the authors, I also would like to thank the reviewers who kindly agreed to evaluate the manuscript.
Below, you will find the answers to the reviewers' comments.
On behalf of all authors,
Response to reviewer 1
- It is not clear why the structures of L-asparaginase were not relaxed by molecular dynamics. It is known that pdb structures are the result of processing X-ray structural analysis data and protein structures were obtained in vacuum. For most in silico work, pdb structures are recommended to be processed by molecular dynamics.
We thank the reviewer for the comments, and we would like to clarify some points. Regarding the quality of structural models, the PDB structures used for E. coli (3ECA) and D. chrysanthemi (2JK0) have resolutions of 2.40 and 2.50 Å, respectively. Also, the R-values are 0.111 and 0.188, indicating well-refined structures. For both structures, hydration shells were resolved with the presence of 465 (0.95 RSCC=Real-Space-Correlation-Coefficient) and 418 (0.91 RSCC) water molecules, respectively, indicating a high agreement between the density map data and the three-dimensional model.
Despite the high quality of the structures of 3ECA and 2JK0, as indicated by the refinement parameters, the reviewer raises an interesting point that should be considered. To address this, we conducted an additional analysis to compare the differences between structure 3ECA and a higher quality structure, 1NNS (Resolution: 1.95 Å, R-value: 0.130). A pairwise structure alignment using rigid-body superposition (jFATCAT-rigid) revealed a 0.27 Å RMSD, leading to the conclusion that these independently resolved structures are essentially identical.
Moreover, the 3ECA structure has been cited in more than 70 papers, including one from April 2024, where it was used for structural comparison to a newly solved structure from Rhodospirillum rubrum (RMSD 1.7 Å; doi:10.1002/pro.4920).
Also, we want to emphasize that the structure models we used have been employed in numerous other studies, many of which did not involve additional molecular dynamics post-processing (e.g., 10.1002/pro.4920). The primary objective of using the PDB structures was to identify the general spatial arrangement of amino acids in solvent-exposed regions, correlating them with immunogenicity and allergenicity. Considering our immediate goals, we believe that further structural processing will not impact our downstream analyses. Therefore, we kindly request the reviewer to reconsider this recommendation.
- The authors should describe in more detail the algorithms for identifying epitopes in association with alleles.
The recommendation was considered and changes to the manuscript were made including the following citation: “The method used was NetMHCIIPan 4.1 EL (release; May 2023), that consists of a neural network that predicts MHC binding values from an amino acid sequence, based on a training set of peptide-MHC class II quantitative binding data covering thou-sands of human MHC molecules, including HLA-A, HLA-B, HLA-C, HLA-E, HLA-G [76].”
- It is also necessary to disclose in more detail the algorithm that made it possible to obtain specific sequences on lines 310-312.
The recommendation was considered and changes to the manuscript were made including the following citation: “Mapping of structure sequences conservation was performed using the UCSF Chimera program. Visualization of sequence conservation onto molecular structures was per-formed using the UCSF Chimera program [75].”
The technical notes were considered.
Reviewer 2 Report
Comments and Suggestions for Authors
The manuscript entitled: “A structural in silico analysis of the immunogenicity of L-asparaginase from Penicillium cerradense” is a well written manuscript that deals with an interesting topic as this of characterizing enzymes to be used as alternatives for therapeutic protocols. However, there are some concerns that need to be addresses before the manuscript is accepted for publication.
· In lines 72-77 the authors refer to the adverse effects of ASNase from bacterial sources. Does this have to do with differences in structure of ASNase from different origin that are studied in this manuscript?
· The authors are advised to add to each paragraph in results section, two sentences to show the purpose of performing the analysis, as well as a conclusion of their findings, where appropriate.
· Line 171: Figure S2 is referred in text before S1. Please name figures according to the order that they first appear in the text.
· There is no legend for Figure 1.
· Lines 236-240: Please refer to the algorithm used to predict the density of epitopes and DI.
· In paragraph 4.6 the authors write how they calculated the relative frequency calculation for epitope density determination. In results section though the also use the term “degree of immunogenicity”. It is not clear what this term stands for and how it was calculated.
· In Figure 6 the legend refers to immunogenicity degree but in axis title relative frequency is shown. In figure 7 also the legend refers to immunogenicity degree, in text the relative frequency and on the figure the epitope density.
· Paragraph 2.5 contains a lot of information regarding degree of immunogenicity, relative frequency etc but there is no, short, comment on the results to highlight the meaning of these results. Please explain the difference between these two terms (DI and relative frequency) and explain why it is necessary to use them both, since the amount of information is confusing.
· Lines 375-383. This paragraph seems to be the conclusion of the current study. Please move it to the appropriate section after Discussion.
· Lines 421-422. Please add a reference that justifies this assumption.
· Most of the references in discussion section are until 2016. Please try to include more recent bibliography.
· In several points in text genus and species names are not in italics (eg lines 132, 159).
Author Response
April 22, 2024
Dear Editors and Reviewers,
Thank you for considering the manuscript as potentially valuable for publication. On behalf of the authors, I also would like to thank the reviewers who kindly agreed to evaluate the manuscript.
Below, you will find the answers to the reviewers' comments.
On behalf of all authors,
Response to reviewer 2
In lines 72-77 the authors refer to the adverse effects of ASNase from bacterial sources. Does this have to do with differences in structure of ASNase from different origin that are studied in this manuscript?
The adverse effects presented in the introduction of this work are adverse effects already highlighted and associated with the clinical use of ASNase from bacterial sources. The references used for such restrictions did not correlate adverse effects on the ASNase structure.
In paragraph 4.6 the authors write how they calculated the relative frequency calculation for epitope density determination. In results section though the also use the term “degree of immunogenicity”. It is not clear what this term stands for and how it was calculated.
The recommendation was considered and changes to the manuscript were made including the following citation: “The prediction of T-cell epitope density allows the inference of the degree of immunogenicity (DI) [42-44]. this measurement alternative conceptualizes and compares a more immunogenic protein to another if the epitope density is greater. In this regard, evaluating the immunogenicity of proteins of therapeutic value has commonly used the density of epitopes as an indicator, inferring the degree of immunogenicity [45-47].”
Lines 421-422. Please add a reference that justifies this assumption.
“ASNase from P. cerradense showed a similar or smaller T-cell/B-cell immunogenicity compared to the E. coli and D. chrysanthemi ASNase. These results may represent a degree of immunogenicity for P. cerradense's ASNase compatible with clinical use.”
The recommendation was considered and changes to the manuscript were made including the following citation:
These results may represent a degree of immunogenicity for P. cerradense's ASNase compatible with clinical use was deleted and a new sentence rewritten.
It might be possible to infer that Penicillium cerradense's ASNase immunologic safety required for clinical use is similar to those already marketed. However, in vitro and/or in vivo evaluation is needed to confirm these assumptions. "
The other technical notes were considered.
Round 2
Reviewer 1 Report
Comments and Suggestions for Authors
Authors answered all my questions. I recommend this article for publication